# Bismuthene for highly efficient carbon dioxide electroreduction reaction

Fa Yang[1,2,5], Ahmed O. Elnabawy [3,5], Roberto Schimmenti[3,5], Ping Song[1], Jiawei Wang[1], Zhangquan Peng[1], Shuang Yao[4], Ruiping Deng[4], Shuyan Song [4], Yue Lin [2], Manos Mavrikakis[3✉] & Weilin Xu [1✉]

Bismuth (Bi) has been known as a highly efficient electrocatalyst for $CO_2$ reduction reaction. Stable free-standing two-dimensional Bi monolayer (Bismuthene) structures have been predicted theoretically, but never realized experimentally. Here, we show the first simple large-scale synthesis of free-standing Bismuthene, to our knowledge, and demonstrate its high electrocatalytic efficiency for formate ($HCOO^-$) formation from $CO_2$ reduction reaction. The catalytic performance is evident by the high Faradaic efficiency (99% at −580 mV vs. Reversible Hydrogen Electrode (RHE)), small onset overpotential (<90 mV) and high durability (no performance decay after 75 h and annealing at 400 °C). Density functional theory calculations show the structure-sensitivity of the $CO_2$ reduction reaction over Bismuthene and thicker nanosheets, suggesting that selective formation of $HCOO^-$ indeed can proceed easily on Bismuthene (111) facet due to the unique compressive strain. This work paves the way for the extensive experimental investigation of Bismuthene in many different fields.

[1] State Key Laboratory of Electroanalytical Chemistry, & Jilin Province Key Laboratory of Low Carbon Chemical Power, Changchun Institute of Applied Chemistry, Chinese Academy of Sciences, 5625 Renmin Street, 130022 Changchun, P.R. China. [2] University of Science and Technology of China, 230026 Anhui, P.R. China. [3] Department of Chemical & Biological Engineering, University of Wisconsin-Madison, Madison, WI 53706, USA. [4] State Key Laboratory of Rare Earth, Changchun Institute of Applied Chemistry, Chinese Academy of Sciences, 5625 Renmin Street, 130022 Changchun, P.R. China. [5] These authors contributed equally: Fa Yang, Ahmed O. Elnabawy, Roberto Schimmenti. ✉email: emavrikakis@wisc.edu; weilinxu@ciac.ac.cn

Electrochemical reduction reaction of $CO_2$ (CO2RR) to useful products provides one practical way to deal with the urgent problems of global warming and energy crises[1,2]. Substantial effort has been devoted to searching for catalysts for such process to produce valuable products[3], such as formic acid (HCOOH) or formate ($HCOO^−$)[4–6].

Over the past three decades, researchers have evaluated a variety of bulk metals (such as Pb, Hg, In, Cd, Sn, Co) as electrodes for CO2RR to produce formate in aqueous solutions[7,8]. Unfortunately, these electrodes usually suffer from high cost, large overpotential, limited availability, poor selectivity as well as poor durability, which seriously hinder their large-scale practical applications[9,10]. To address these issues, several electrocatalytic systems based on naturally abundant and chemically stable metal elements have been recently explored as potential $CO_2$ electroreduction catalysts[2,11,12]. Bismuth (Bi), typically used as CO2RR electrocatalyst, has advantages over other traditional metals due to its low toxicity, cost-effectiveness, and higher stability. However, previously reported bismuth electrocatalysts for CO2RR usually require large overpotentials or only present low current densities. For example, metal Bi, Bi single atomic[13], Bi spherical nanoparticles[14,15], Bi nanobelts[16], thin films[17], Bi nanowires[18], and $Bi_2O_3$ nanotubes[11] have been synthesized, but all with very limited CO2RR catalytic performance to $HCOO^−$ production. Two-dimensional (2D) multilayered Bi nanosheets with limited CO2RR performance[19–21] have also been synthesized recently with state-of-the-art synthesis methodologies, including, but not limited to, sonochemical exfoliation of bulk Bi[22] and in situ topotactic electroreduction of Bi precursors[23]. The monolayer Bi nanosheet (Bismuthene) was experimentally realized recently over templates such as SiC[20] and $NbSe_2$[24], while the unsupported, stable, free-standing Bismuthene has never been synthesized before, but predicted and studied theoretically in the context of investigating its thermoelectric and other electronic properties[25–28].

Here we report a simple, scalable, wet chemical method to synthesize stable free-standing hexagonal Bismuthene and demonstrate its superior electrocatalytic CO2RR performance towards $HCOO^−$ production. Density functional theory (DFT) calculations suggest that this high catalytic performance can be attributed to the large compressive strain found in free-standing Bismuthene.

## Results

### Synthesis and characterization of Bismuthene

Based on a simple reduction reaction with Bismuth (III) chloride ($BiCl_3$) as the Bi precursor and $NaBH_4$ as the reductant (see Methods for details, Supplementary Information), a series of free-standing 2D bismuth nanosheets (BiNSs) with different thicknesses (from monolayer Bismuthene to multilayers) were synthesized (Supplementary Figs. 1, 2). Figure 1a–c shows the typical transmission electron microscopy (TEM) images of the thinnest BiNSs (Bismuthene) with the (111) facet exposed (Fig. 1d, e, Supplementary Fig. 3 and Supplementary Note 1), while the thicker BiNSs have the (011) facet exposed (Supplementary Fig. 4) probably due to the temperature difference of synthesis[29,30]. Typical atomic force microscopy (AFM) image shows clearly an average thickness of 0.65 nm of Bismuthene (Fig. 1f, g and Supplementary Fig. 5), consistent with the measurement from lateral high-angle annular dark-field scanning transmission electron microscopy (HAADF-STEM) image (Fig. 1h), which clearly confirms its single-atom layer thickness with a zig-zag structure[2,31,32]. X-ray photoelectron spectroscopy (XPS) for Bismuthene (Fig. 2a) demonstrates the existence of metallic Bi (0), as indicated by the peaks at 157.1 and 162.7 eV[33]. Moreover, as shown in Fig. 2b, the two Raman bands

at 74 and 99.7 $cm^{−1}$, characterizing the Eg and A1g modes of metallic Bi (0), respectively[34], appear at higher wavenumbers than other thick BiNSs[10], further confirming the metallic state of Bi atoms in Bismuthene. All these results indicate that the obtained metallic Bismuthene can remain stable at ambient conditions. Energy-dispersive X-ray spectroscopy (EDS) on such single metallic Bismuthene sheet shows no residual solvent or surfactants on the surface of Bismuthene nanosheets (Supplementary Fig. 6). Significantly, it was further found that the morphology and thickness of the monolayer sheets can be maintained even after being oxidized at 400 °C in air (Supplementary Figs. 4b, c, d, 7), indicating the ultrastability of such Bi single-atom-thick layers.

### Thickness-dependent CO2RR performance of Bi nanosheets

As for the thickness-dependent catalytic performance of BiNSs for CO2RR, Fig. 3a shows that, with the thickness decrease of the Bi nanosheets, both the current and the onset potential increase correspondingly, apparently indicating that for CO2RR on BiNSs, the thinner the better. The much better performance of Bismuthene with (111) facet exposed compared to thick BiNSs with (011) facet exposed could be due to the different activity of different exposed facets. Additionally, the slightly better performance of 4.2-nm BiNSs than the 11.3-nm BiNSs could be partially attributed to the larger surface area of the thinner nanosheets[35].

Furthermore, as shown in Supplementary Fig. 8, $H_2$ from hydrogen evolution reaction (HER) is the only gas product detected from gas chromatography (GC) analysis, and formate ($HCOO^−$) detected in solution from $^1H$ nuclear magnetic resonance (NMR) analysis is the only product from CO2RR. Consistent with the current data shown in Fig. 3a, the partial current density ($j_{HCOO^−}$, Supplementary Fig. 9a) and the Faradaic efficiency ($FE_{HCOO^−}$, Fig. 3b) for formate production from CO2RR increase with the thickness decrease of BiNSs for a given applied potential. Significantly, Bismuthene shows the best performance with high selectivity (~100% for formate formation) and $FE_{HCOO−}$ (up to 98%) at a potential of −580 mV (Supplementary Table 1) as well as an ultrasmall onset overpotential of <90 mV (Supplementary Fig. 9b)[36]. To understand such observed thickness-dependence of CO2RR performance of BiNSs, the electrochemically active surface areas (ECSAs) (Fig. 3c and Supplementary Fig. 9c), the $CO_2$ adsorption isotherms (Supplementary Fig. 10), and the $CO_2$ temperature-programmed desorption ($CO_2$-TPD, Supplementary Fig. 11 and Supplementary Note 2) of these nanosheets were all determined. All these results are consistent with the thickness-dependent performance shown in Fig. 3a, indicating that the higher $CO_2$ adsorption capability and higher CO2RR activity of (111) facet exposed on Bismuthene sheets could be partially attributed to the much larger amount of active sites on Bismuthene sheets than on thick BiNSs[37].

To better understand the electrochemical activity and kinetics of CO2RR on BiNSs, Tafel analysis was performed on the $HCOO^−$ partial current density. As shown in Fig. 3d, the Tafel slope for $HCOO^−$ formation on 0.65 nm BiNSs (Bismuthene) yielded a value of 87.6 mV $dec^{−1}$, much smaller than the values obtained from the thick BiNSs with thickness of 4.2 nm (115.4 mV $dec^{−1}$) and 11.3 nm (127.6 mV $dec^{−1}$). The much smaller Tafel slope of the 0.65 nm Bismuthene than the thicker ones indicates a much faster electron transfer rate on Bismuthene, consistent with the Nyquist plots shown in Fig. 3e and Supplementary Fig. 12. Further experiment (Supplementary Fig. 13) indicates that the concentration of $HCO_3^−$ can influence the efficiency of CO2RR on Bismuthene with reaction order of about 0.98[38].

We further examined the durability of Bismuthene for long-term CO2RR by collecting the current density (Supplementary Fig. 14) and $FE_{HCOO^−}$ (Fig. 3f) at different potentials. Typically, for

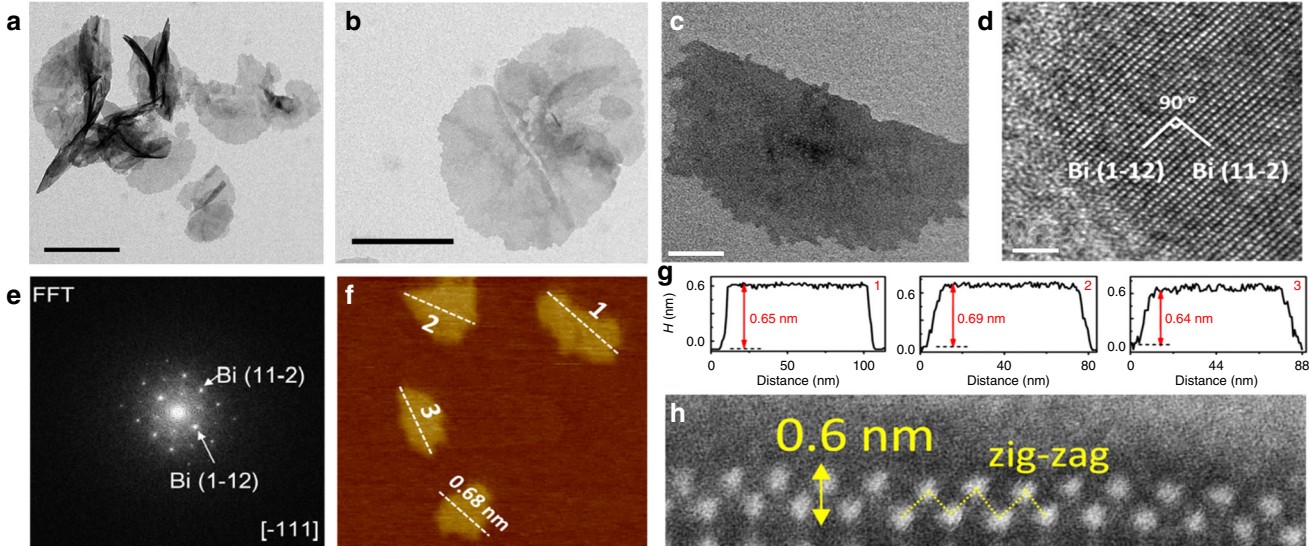

**Fig. 1 Characterization of monolayer Bi nanosheets (Bismuthene). a−c** Typical TEM images. Scale bar in (**a**): 500 nm; scale bar in (**b**): 200 nm; scale bar in (**c**): 50 nm. **d** Typical HRTEM image; scale bar: 2 nm. **e** FFT of (**d**). **f** Typical atomic force microscopy (AFM) image. **g** The corresponding height profiles for three Bismuthene nanosheets marked in (**f**). **h** Typical lateral HAADF-STEM image of a Bismuthene nanosheet, directly showing the single-atom thickness of the layer with zig-zag structure.

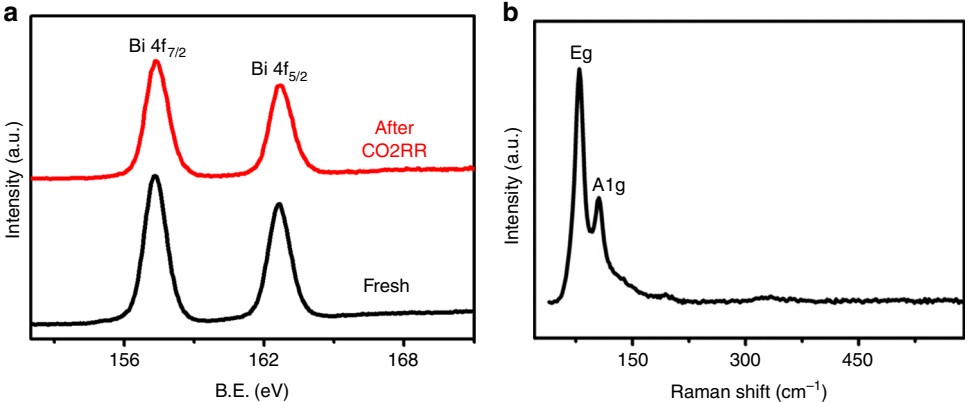

**Fig. 2 Physical characterization of Bismuthene. a** High-resolution Bi 4f XPS spectra before and after 75-h CO2RR process at −0.58 V. **b** Raman spectrum of fresh metallic Bismuthene nanosheets.

Bismuthene nanosheets at −0.58 V, as shown in Fig. 3f, in a long time (75 h) window, both the current density and the corresponding $FE_{HCOO^-}$ were maintained steadily, indicating a remarkable durability of Bismuthene nanosheets for CO2RR. Significantly, Supplementary Figs. 15, 16 show that the thinner the BiNSs are, the better their long-term durability for CO2RR. Consequently, Bismuthene shows the best durability among these BiNSs with different thicknesses. Such remarkable durability of Bismuthene nanosheets could be due to the quicker electron transport on the 2D monolayer Bismuthene (Fig. 3e), which then leads to smaller effect and hence guarantees the remarkable long-term durability in aqueous electrolyte[39]. The exceptional long-term durability of Bismuthene during the CO2RR was further confirmed unambiguously by the minor variation of XPS spectrum (Fig. 2a) and ECSA (Supplementary Fig. 9d), Raman spectra shown in Supplementary Figs. 17, 18, morphology and thickness of Bismuthene nanosheets (Supplementary Note 3 and Supplementary Figs. 19–21). Further, this durability of performance is consistent with the observation of no morphology variation (Supplementary Fig. 5) and no CO2RR performance decay (Supplementary Fig. 22) of Bismuthene after annealing at 400 °C in air.

We have shown that pure Bismuthene nanosheets are highly efficient towards the $HCOO^-$ formation process from CO2RR. Nonetheless, due to the stacking of layered-Bismuthene, a compact catalyst layer (Fig. 4a) was formed on electrode. In such compact layer, many active sites on Bismuthene nanosheets are not available to reactants, thus leading to very limited apparent current (Supplementary Fig. 23) or partial current density ($j_{HCOO^-}$) for $HCOO^-$ (Fig. 4b) formation. Such fact then further limits the large-scale applicability of pure Bismuthene[5]. To solve this problem, the Bismuthene nanosheets were mixed with a small amount of inert carbon black (BP2000, noted as BP, which is inactive to CO2RR in the potential range studied here (Supplementary Fig. 24)) to inhibit the compact stacking of Bismuthene nanosheets, leading to the formation of a non-compact catalyst layer (Fig. 4c) with ECSA increased by more than three times (Fig. 4d). The $j_{HCOO^-}$ (up to 54 mA cm$^{-2}$, Fig. 4b) and the product formation rate of $HCOO^-$ (Supplementary Fig. 23b) obtained on such new catalyst layer (Bismuthene@BP, with BP 3 wt.% optimally) are more than eight and four times of that obtained on pure Bismuthene, respectively. Such BP-induced improvement of CO2RR efficiency

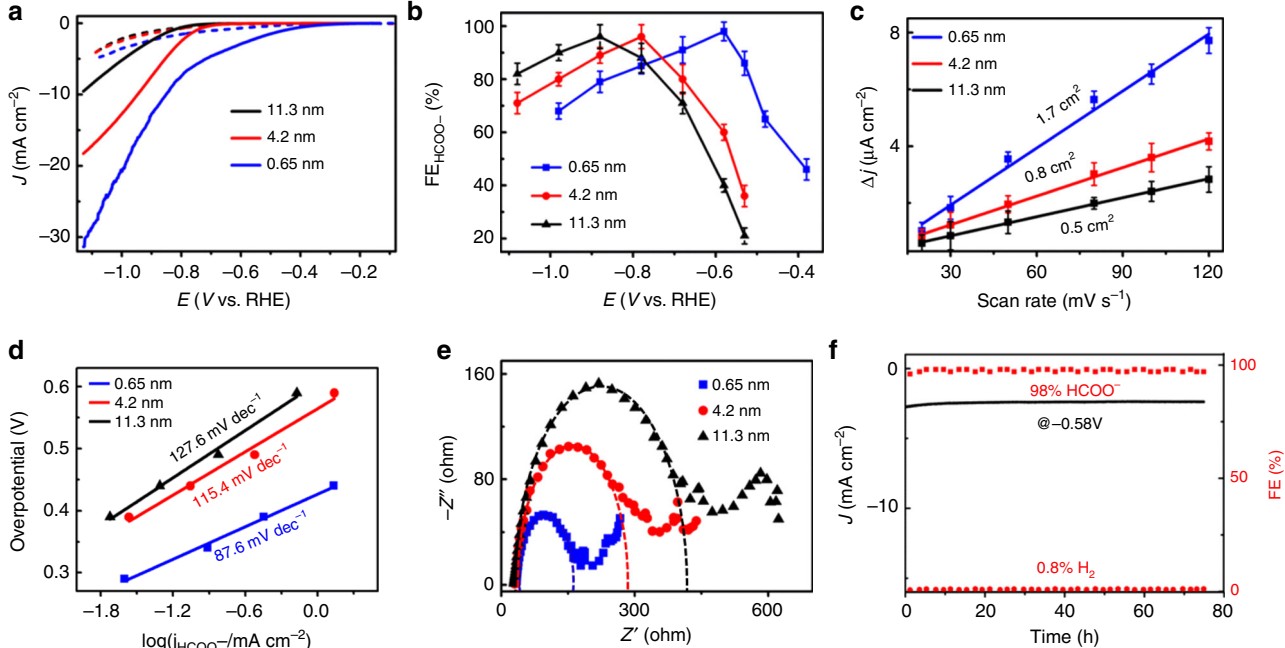

**Fig. 3 Electroreduction of $CO_2$ to formic acid on BiNSs. a** pH-corrected linear sweep voltammetric curves (LSV) in the $CO_2$ saturated (solid line pH 7.2) and $N_2$ saturated (dash line pH 8.8) 0.5 M $KHCO_3$ aqueous solution with the same Bi loading of 0.39 mg/$cm^2$ on glassy carbon electrode. **b** Comparison of Faradaic efficiencies for formate at each applied potential for BiNSs with different thicknesses. **c** Charging current density differences $\Delta j$ plotted against scan rates on BiNSs with different thickness for the calculation of ECSA. **d** Tafel plots of the partial $HCOO^-$ current density for BiNSs with different thicknesses. **e** Nyquist plots for BiNSs with different thicknesses. The dotted curves are the fittings. **f** Long-term stability of Bismuthene nanosheets (0.65 nm) at a potential of −0.58 V and the corresponding FEs for $HCOO^-$ and $H_2$. All the error bars in (**b**, **c**) represent the standard error of the mean.

was also observed on thicker BiNSs as shown in Supplementary Figs. 23c, d, 25, but not as much as that observed on Bismuthene (Fig. 4). These greatly improved catalytic properties, along with the maintained superhigh $HCOO^-$ selectivity (~100% with $FE_{HCOO^-}$ ~ 99%) and remarkable long-term durability (>75 h) (Supplementary Fig. 26), make such Bismuthene indeed practical for potential large-scale application. In all, such improvement of CO2RR performance could be attributed in part to the BP-induced variations of morphology (compact and noncompact), Bi content and ECSA of the catalyst layer and then the variation of mass transportation in it.

To summarize, we compare Bismuthene to an extensive list of other reported catalysts for CO2RR to formate (see Supplementary Table 1). Bismuthene offers a superhigh $FE_{HCOO^-}$ (99% at a small potential of −580 mV) and durability (no performance decay after 75 h and annealing at 400 °C)[40]. Therefore, Bismuthene is a very promising CO2RR electrocatalyst according to activity, stability, and selectivity to $HCOO^-$.

**Theoretical understanding to CO2RR performance on Bi nanosheets**. Density functional theory calculations were employed to determine the origin of the remarkable electrocatalytic properties of Bismuthene for CO2RR. In particular, the structure-sensitivity of the CO2RR mechanism was studied on Bismuthene monolayers and on thicker nanosheets models. According to the TEM analysis, the obtained Bismuthene monolayers expose the (111) facet (Fig. 1d, e), while thicker nanosheets expose the (011) facet (Supplementary Fig. 4). Thus, we calculated the most stable adsorption structures and energetics of OCHO∗ (formate), COOH∗ (carboxyl), CO∗, and OH∗ for CO2RR, as well as H∗ for the competing HER, on a Bismuthene monolayer exposing the (111) facet and on a thicker nanosheet model, exposing the (011) facet (see Supplementary Information

and Supplementary Fig. 27 for details on the construction of these models and the respective optimized lattice constants). The free energy diagrams of CO2RR and HER on these model surfaces are reported in Fig. 5a, b, respectively (Supplementary Table 2 lists enthalpic, entropic and solvation energy contributions to the calculated Gibbs free energy).

On Bismuthene (111) (see Fig. 5a), HER has an overpotential of 1.28 V, whereas CO2RR has an overpotential (through the OCHO∗ path) of only 0.58 V, which is reasonably close to the experimental observation (Fig. 3). As a result, CO2RR is preferred to HER. This explains the experimentally determined high Faradaic efficiency of Bismuthene towards $HCOO^-$. Conversely, the mechanistic picture is quite different for the (011) thick Bi nanosheet model; see Fig. 5b. OCHO∗ is very stabilized on that surface, to the extent that its reduction to HCOOH is endergonic. Similar to the Bismuthene (111) model, COOH∗ is significantly less stable than OCHO∗ on the (011) model, but this makes COOH∗ the more viable path to formic acid for this strongly adsorbing surface. It is thermodynamically much more likely, however, that COOH∗ would dissociate to produce the highly stable CO∗ + OH∗ state, rather than produce HCOOH. This points out to the likelihood of a (011) surface that is heavily covered with reaction intermediates, specifically OCHO∗ and OH∗ which are more stable than CO∗ and COOH∗. Higher coverage of these reaction intermediates would likely destabilize adsorbates[41,42], pushing the entire free energy diagram up to higher energies, including that of HER. The adsorption structures shown in Fig. 5b and Supplementary Figs. 28, 29 explain the origin of the strong adsorption on the (011) thick BiNSs, on which adsorption is exclusive to the step edge sites, indicating that those step edges tend to be poisoned by reaction intermediates, leaving the terraces as the only available sites for electrocatalysis. Nevertheless, the terraces of (011) thick BiNSs also bind adsorbates stronger than the (111) Bismuthene model

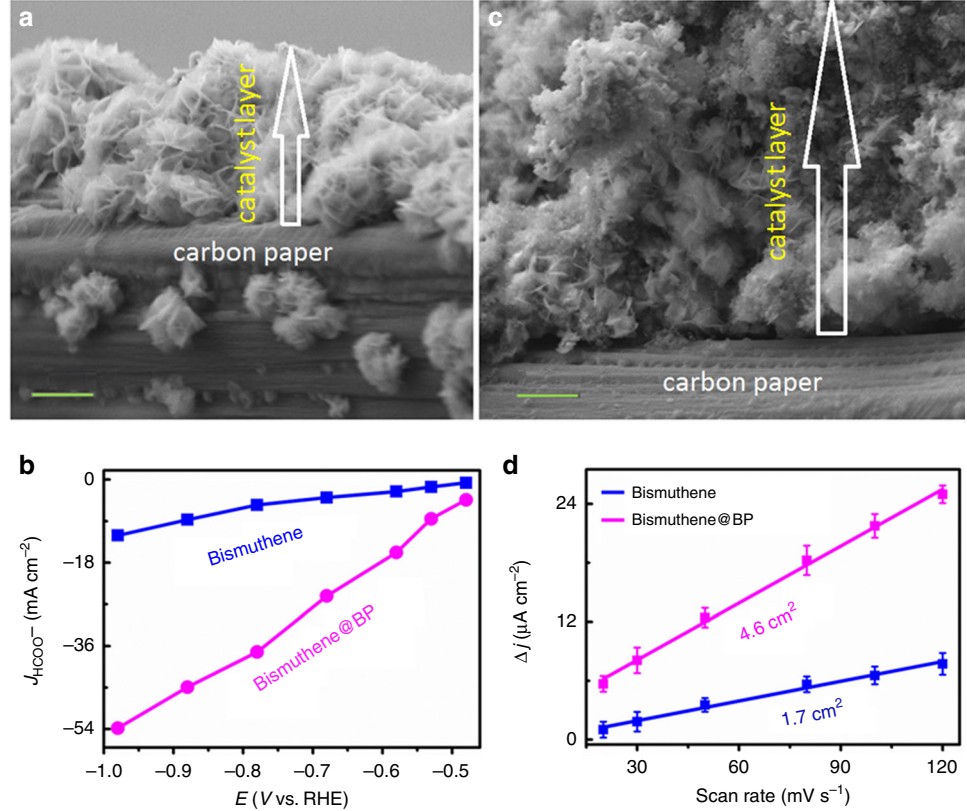

**Fig. 4 BP-induced enhancement of CO2RR performance on Bismuthene. a** SEM image of the transect to show the compact catalyst layer of Bismuthene nanosheets on carbon paper electrode. **b** Partial current density for HCOO⁻ ($j_{HCOO^-}$) versus potential on Bismuthene and Bismuthene@BP (with BP 3 wt.% optimally). **c** SEM image of the transect to show the noncompact catalyst layer of Bismuthene@BP on carbon paper electrode. The arrows in (**a**, **c**) indicate the thickness difference of the catalyst layers with the same mass loading, scale bars in (**a**, **c**): 1.0 μm. **d** ECSA measurement for both pure Bismuthene and Bismuthene@BP. All the error bars in (**d**) represent the standard error of the mean.

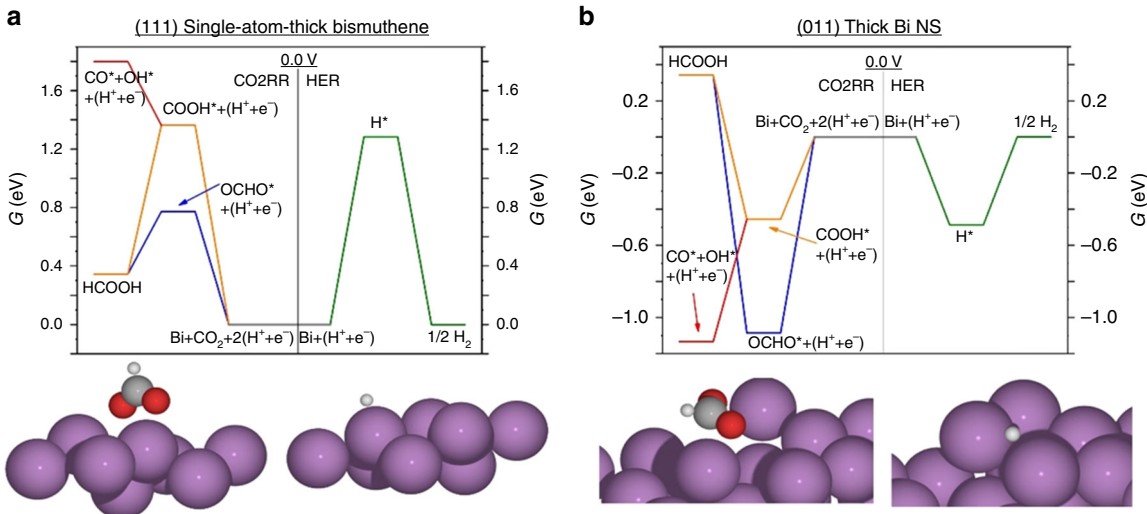

**Fig. 5 DFT calculations. a**, **b** Calculated free energy diagrams for CO2RR and HER on (111) single-atom-thick Bismuthene (**a**) and (011) thick Bi nanosheets (**b**) at 0.0 V. HER is represented in green, and CO2RR through OCHO* and COOH* are represented in blue and orange, respectively. The state of CO* + OH* is represented in red. The insets below the diagrams depict the adsorption structures of OCHO* and H*. Violet, red, gray, and white spheres in the insets represent Bi, O, C, and H atoms, respectively.

(Supplementary Figs. 30, 31), thus hindering the activity of all sites on the thick Bi NSs compared to the thin Bismuthene (111). The weaker adsorption on Bismuthene could be attributed to the compressive strain the ultrathin nanosheet model experiences (our calculations reveal up to 6% compression relative to bulk Bi structure; see Supplementary Information for details). Surface compressive strain has been known to weaken adsorption on most surfaces, and this is especially relevant for ultrathin structures and two-dimensional materials, for which the thinner, the more compressive the strain, and the weaker the adsorption

on them[43]. This explains our experimental trends (Supplementary Figs. 15, 16), in which the thicker nanosheets would become more prone to deactivation during long-term CO2RR process, resulting in low activity for CO2RR. We note that the stronger adsorption on the thick NSs predicted by our calculations is consistent with our experimental $CO_2$-TPD (Supplementary Fig. 11), in which the $CO_2$ desorption from thick BiNSs required higher temperature, relative to desorption from single-atom-thick Bismuthene. Therefore, Bismuthene is a promising candidate for selective CO2RR to formate, maintaining remarkable stability.

Moreover, as shown in Fig. 5, the DFT results predict that, without the competition of CO2RR, the intrinsic HER activity of thick BiNSs is much higher than that of Bismuthene as indicated by the much smaller energy barrier. To verify this prediction, we studied the HER activity of these BiNSs. In this experiment, the Ar-saturated 0.5 M $KH_2PO_4/K_2HPO_4$ (pH 7.0) solution rather than $KHCO_3$ solution was adopted to avoid any possible effect of CO2RR process on the HER activity of these BiNSs. Interestingly, as shown in Supplementary Fig. 32, the intrinsic HER activity of the thick BiNSs is indeed much higher than that of Bismuthene as indicated by the onset potential of HER, in agreement with our DFT results.

## Discussion

In conclusion, Bi single-atom layers were successfully synthesized for the first time. Owing to the very high atom utilization, high surface density of intrinsically more active sites, high electronic conductivity, and superior structural stability of atomic layers, Bi single-atom-thick layers are capable of selectively catalyzing the electroreduction of $CO_2$ to $HCOO^-$ only. The catalytic performance is underlined by the high Faradaic efficiency of ~99%, high durability (>75 h) and low onset overpotential of <90 mV. Our mechanistic analysis shows that the thicker BiNSs, exposing the (011) facet, bind reaction intermediates very strongly, likely leading to their poisoning by those species. This explains their lower activity and stability relative to the atomically thin Bismuthene. The latter, exposing the (111) facet, shows low overpotential for CO2RR, through the OCHO∗-mediated mechanism, and prohibitively high overpotential for HER, in agreement with experiments. This work provides a simple synthesis for the—until now—synthetically elusive Bismuthene, and proves that this material outperforms others related materials in activity and stability, thus taking us one step closer towards the efficient conversion of $CO_2$ to liquid fuels.

## Methods

**Chemicals**. Bismuth (III) chloride ($BiCl_3$) was obtained from damas-beta, 2-ethoxyethanol ($C_4H_{10}O_2$) was obtained from Tianjin Fuchen Factory, sodium borohydride ($NaBH_4$) was obtained from Sigma-Aldrich, potassium bicarbonate ($KHCO_3$) was obtained from Beijing Chemical Work, and the Nafion solution (5 wt%) was obtained from Sigma-Aldrich. All chemicals were used as delivered without further treatment. Carbon paper was purchased from CeTech Co., Ltd. Platinum wire (99.997%) was purchased from Premion. All reagents were of analytical grade and used without further purification. Deionized water (resistivity of 18.2 MΩ cm) was used in all solution preparations. Nitrogen ($N_2$, 99.999%) and carbon dioxide ($CO_2$, 99.999%) were purchased from Junyang Co., Ltd.

**Instrumentation**. High-resolution transmission electron microscopy (HRTEM) was performed using a JEOL JEM-2100 electron microscope with an operating voltage of 200 kV. HAADF-STEM images were obtained on a JEOL LEM 2200FS/TEM, equipped with a CEOS probe corrector. AFM measurements were performed with a Veeco Multimodel Nano Scope 3D in the tapping AFM mode. X-ray diffraction measurements were performed using a Rigaku-D/MAX-PC 2500 X-ray powder diffractometer with Cu $K\alpha$ X-ray source. Photoelectron spectroscopic (XPS) measurements were performed on an AXIS Ultra DLD (Kratos company) using a monochromic Al X-ray source. The Raman spectrum was obtained on a laser confocal Raman spectroscopy (Labram-010, Horiba-JY) employing the Nd: YAG laser wavelength of 633 nm. [1]H-NMR was performed on a BRUKER AVANCE-III 500 HD (Switzerland). Electrochemical experiments were performed using a CHI 750E electrochemical work station (CH Instruments, Chenhua Co., Shanghai, China).

**Preparation of BiNSs with an average thickness of 0.6 nm**. Bismuth (III) chloride (1.5 mmol) was dissolved into 50 mL 2-ethoxyethanol, followed by ultrasonicating the mixture to form a uniform and transparent solution. After vigorous stirring for 30 min with bubbling argon in an oil bath at 120 °C and cooling naturally to room temperature, 20 mL aqueous solution of $NaBH_4$ (60 mmol) was added dropwise to the bismuth solution under argon atmosphere and the mixture was stirred for 5 min at ambient temperature. The resultant black precipitate was collected by filtration and washed with water and ethanol twice, respectively. The obtained powders were dried in a vacuum oven at room temperature and then stored in sealed Ar atmosphere at 4 °C for further characterization or application.

**Electrochemical measurements**. The carbon ink was formed by mixing 5 mg of BiNSs, 50 μL of 5 wt% Nafion solutions, and 950 μL of ethanol together to form a homogeneous ink by ultrasonication. A 10 μL aliquot of the ink was dropped on the surface of the glassy carbon rotating disk electrode (4 mm in diameter, and its apparent surface area is 0.1256 cm²), linear sweep voltammetry was performed from −0.8 to −1.8 V (vs. SCE) at a scan rate of 5 mV s⁻¹ after purging the electrolyte with $N_2$ or $CO_2$ gas for 30 min. The electrolyte was 0.5 M $KHCO_3$ solution. A platinum wire and a saturated calomel electrode SCE (with saturated KCl as the filling solution) were used as counter and reference electrodes, respectively. For the detection of electrocatalytic reduction of $CO_2$, 200 μL of homogeneous ink (1 mg catalyst) was evenly drop-cast onto a carbon paper (1 × 1 cm²) surface.

**Electrocatalytic reduction of $CO_2$**. All the experiments were carried out on a three-electrode system with a gas-tight two-compartment H-cell. The two compartments separate working and counter electrodes with a cation-exchange membrane (Nafion 117). Electrode potentials were converted to the reversible hydrogen electrode (RHE) reference scale using $E_{RHE} = E_{SCE} + 0.242 + 0.0591 \times$ pH. Each compartment holds 50 mL of electrolyte. The electrolyte was 0.5 M $KHCO_3$ saturated with $CO_2$ with pH of 7.2. Before the experiments, the electrolyte in the cathodic compartment was saturated with $CO_2$ by bubbling $CO_2$ gas for at least 30 min. During the reduction experiments, $CO_2$ gas was delivered at an average rate of 30 mL min⁻¹ (at room temperature and ambient pressure) and routed directly into the gas sampling loop of a gas chromatograph (Thermo Trace 1300). The gas chromatograph was equipped with a Molecular Sieve 5A capillary column and a packed Carboxen-10000 column. Helium (99.999%) was used as the carrier gas. The gas chromatograph columns led directly to a thermal conductivity detector to quantify hydrogen and a flame ionization detector equipped with a methanizer to quantify carbon monoxide. The GC was calibrated using different concentrations of calibration standards commercially available from YJ Technical Company.

The Faradaic efficiency (FE) and the partial current densities of $H_2$ production were calculated as below:

$$FE_s = \frac{2Fv_sGp_0}{RT_0i_{total}} \times 100\%,$$

$$j_{H_2} = FE_{H_2} \times i_{total} \times (electrode\ area)^{-1},$$

where $v_s$ (vol%) = volume concentration of $s = H_2$ in the exhaust gas from the electrochemical cell (GC data), $p_0 = 1.013$ bar and $T_0 = 298.15$ K, gas flow rate ($G$) measured by a FL-1802 rotor meter at the exit of the electrochemical cell (mL min⁻¹), $i_{total}$ (mA) = steady-state cell current, $F = 96485$ C mol⁻¹, $R = 8.314$ J mol⁻¹ K⁻¹.

The total amount of liquid product was measured using NMR (AV 500) spectroscopy, in which 0.5 mL electrolyte was mixed with 0.1 ml $D_2O$ (deuterated water), and 0.05 μL dimethyl sulfoxide (DMSO, Sigma, 99.99%) was added as an internal standard. Assuming that two electrons are needed to produce one formate molecule, the FE can be calculated as follows: $FE_{HCOO^-} = 2\ F \times n_{HCOO^-}/(i_{total} \times t)$, where F is the Faraday constant and $t$ is the time, then

$$j_{HCOO^-} = FE_{HCOO^-} \times i_{total} \times (electrode\ area)^{-1}.$$

**ECSA measurement**. ECSA = $R_fS$, in which $S$ stands for the real surface area of the smooth metal electrode, which was generally equal to the geometric area of carbon paper electrode (in this work, $S = 1.0$ cm²). The roughness factor $R_f$ was estimated from the ratio of double-layer capacitance $C_{dl}$ for the working electrode and the corresponding smooth metal electrode (assuming that the average double-layer capacitance is 20 μF cm⁻² for a smooth metal surface and 21 μF cm⁻² for carbon supported metal, that is, $R_f = C_{dl}/(20\ \mu F\ cm^{-2})$ for pure metal and $R_f = C_{dl}/(21\ \mu F\ cm^{-2})$ for BP supported metal. The $C_{dl}$ was determined by measuring the capacitive current associated with double-layer charging from the scan-rate dependence of cyclic voltammetric stripping. For this, the potential window of cyclic voltammetric stripping was 0.3−0.4 V versus RHE (0.5 M $KHCO_3$ solution). The scan rates were 20, 30, 50, 80, 100 and 120 mV s⁻¹. The $C_{dl}$ was estimated by plotting the Δ$j$ = $(j_a − j_c)$ at 0.35 V (where $j_c$ and $j_a$ are the cathodic and anodic current densities, respectively) versus SCE against the scan rate, in which the slope was twice that of $C_{dl}$.

**Morphology of Bi nanosheets before and after electrolysis**. Firstly, 10 µL Bi nanosheets ink with 1 mg Bi mL$^{-1}$ was dropped on Cu-grid-supported carbon film and dried at room temperature during vacuum environment. Then TEM images were taken around some special marks so that we can find these areas exactly after the CO2RR experiments. After that, the Cu-grid-supported carbon film was kept in a three-electrode sealed micro channel with outlet and inlet; CO$_2$-saturated 0.5 M KHCO$_3$ was injected into the tube at a speed of 30 mL min$^{-1}$. Controlled potential electrolysis was performed under certain applied potentials. After that, the Cu-grid was washed gently with small amount of water and TEM images were taken on the same locations based on the special marks found before on the sample grid.

**Calibration analysis of the products from CO2RR**. In the present work, the concentrations of liquid products (HCOO$^-$) were determined from NMR spectrometer with added dimethyl sulfoxide (DMSO) as an internal standard. Here, in order to further evaluate the reliability of such traditional method with DMSO as an internal standard, we obtained the multiple-point calibration curves for liquid product (HCOO$^-$). Simultaneously, the concentration of gas products (H$_2$) was obtained based on multiple groups of calibration standards (different concentrations).

**DFT calculations**. Periodic DFT calculations were performed using the Vienna Ab Initio Simulation Package (VASP)[44,45]. Projector augmented wave (PAW) potentials were used to describe the core electron interactions. PAW potentials—as generated and provided by VASP—account for scalar-relativistic treatment of the core electrons. For Bi, $d$ states were treated as valence states. A plane wave basis set with a kinetic energy cutoff of 400 eV was used to expand the electronic wave function of the valence electrons. The generalized gradient approximation (GGA) using the Perdew, Burke, and Ernzerhof (PBE)[46] exchange-correlation functional was used. Dispersion interactions were accounted through the Tkatchenko and Scheffler (TS) method[47], which has been previously evaluated for the reproduction of van der Waals forces bonding multilayers of pnictogens atoms, such as in phosphorene[48]. This choice is consistent also with recent literature exploring atomic and molecular adsorption on Bi model surfaces[49]. The tetrahedron method with Blöchl corrections for determination of partial occupancy was used. A 16 × 16 × 16 Monkhorst-Pack[50] $k$-point sampling was used for bulk lattice constants optimization, while a 6 × 6 × 1 $k$-point sampling was used for studying the CO2RR mechanism on Bi surface slabs. In order to avoid artifacts in the surface calculations arising from the interactions of periodic replicas in the $z$ direction, we provided for at least 15 Å of vacuum layers in the supercell. All the structures were fully relaxed until the residual Hellmann−Feynman forces acting on each atom were less than 0.01 eV Å$^{-1}$. With this setting, the calculated bulk Bi optimized lattice constants are $a = 4.588$ Å and $c/a = 2.495$, in agreement with experiments[51]. As suggested by SAED measurements, the Bi monolayer exposes a (111) facet, while thicker Bi nanosheets expose a (011) facet. Therefore, for the monolayer system, we used a free-standing Bi(111) monolayer model, showing a hexagonal symmetry and characterized by an optimized lattice constant of 4.304 Å. This lattice constant corresponds to a ~6% compression with respect to that of the bulk. Starting from the optimized monolayer unit cell, a 2 × 2 supercell was generated (see Supplementary Fig. 27a) and was used for studying the CO2RR mechanism. For this set of calculations, all the atoms were allowed to relax and a 6 × 6 × 1 Monkhorst-Pack $k$-point sampling was used. The (011) Bi nanosheets were modeled with a six-layer-thick slab, cut from the optimized bulk Bi structure (see Supplementary Fig. 27b). In this case only the adsorbates and the top three layers of the slab were allowed to relax. The most stable adsorption geometries of OCHO∗, COOH∗, OH∗, CO∗, and H∗ on the Bi(111) monolayer and Bi(011) surface are shown in Supplementary Figs. 28, 29, respectively.

Spin−orbit coupling (SOC) was not included in the calculations as they should not affect quantities derived from differences in energy. We tested the accuracy of our computational settings and the validity of our assumption on SOC by comparing our results with earlier studies[49], who use comparable computational settings, but accounting for SOC. On a Bi (111) extended surface model, the difference in energy between OCHO∗ and COOH∗, a key descriptor for establishing the competition between formate- and carboxyl-mediated CO2RR reaction pathway, is within 0.05 eV. Spin-polarization had a negligible effect on the calculated total energy. All systems had ground state with zero net magnetic moment.

To plot the free energy diagram associated with CO2RR and HER, and calculate their respective overpotentials, we implement the computational reversible hydrogen electrode formalism[52]. Essentially, the potential of 0.0 V is defined for protons and electrons at thermodynamic equilibrium with hydrogen gas at all pH, and at standard conditions of temperature (298.15 K) and pressure (1 atm). The free energy of protons and electrons are adjusted by −|e|U for applied potential, U. Hence, the free energy change of an elementary step in which protons and electrons are consumed (i.e. electroreduction) becomes less endergonic at negative applied bias.

The overpotential is calculated as the equilibrium potential minus the onset potential. The onset potential is the minimum bias to be applied for the entire reaction pathway to be downhill in free energy. The equilibrium potential for the reaction CO$_2$ + 2(H$^+$+e$^-$) → HCOOH is calculated to be −0.17 V, in good agreement with the theoretical equilibrium potential of −0.20 V.

The Gibbs free energy (G) of adsorbed and gas-phase species was calculated as:

$$G = E_{Tot} + ZPE − T^*S,$$

where $E_{Tot}$ is the total energy of the system calculated from DFT, ZPE is the calculated zero-point energy, $S$ is the calculated entropy, and $T$ is the standard temperature of 298.15 K. For the construction of CO2RR free energy diagrams, all the species are referenced to the Bi + CO$_2$ $_{(g)}$ + 2(H$^+$ + e$^-$) state, where Bi is the energy of the clean (111) or (011) Bi slab models and CO$_2$ $_{(g)}$ is the gas-phase Gibbs free energy of CO$_2$. Solvation energy contributions to G were calculated using the implicit solvation model as implemented in VASPsol[53]. Additionally, we adopted the CO$_2$ gas-phase PBE energy correction of +0.17 eV previously suggested [54], as the Tkatchenko−Scheffler dispersion method we applied does not affect this relative, surface-independent correction.

Binding energies (BE) of OCHO∗, COOH∗, OH∗, CO∗, and H∗ on the Bi(111) single-atom-thick monolayer and Bi(011) slab model were calculated as follows:

$$BE = E_{tot} − E_{Bi} − E_{gas},$$

where $E_{tot}$ is the total energy of the Bi slab + adsorbate system, $E_{Bi}$ is the energy of the clean Bi(111) single-atom-thick monolayer or Bi(001) slab models, and $E_{gas}$ is the gas-phase energy of the adsorbate. According to this definition, more negative BE indicates stronger adsorption.

## Data availability

Additional data are provided in the Supplementary Information. All the data that support the findings of this study are available from the corresponding authors upon reasonable request.

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

## Acknowledgements

This work was funded by the National Natural Science Foundation of China (21925205, U1601211, 21633008, 2017YFE9127900, 21733004, 21721003, 2018YFB1502302, and 21972133), K.C. Wong Education Foundation and Science and Technology Innovation Foundation of Jilin Province for Talents Cultivation (20160519005JH, 20170414019JH) and Jilin Youth foundation (20160520137JH). The work at UW-Madison was supported by the Paul A. Elfers Professorship and WARF Named Professorship funds; the computational work was performed using supercomputer resources at National Energy Research Scientific Computing Center (NERSC), supported by the U.S. DOE, Office of Science under Contract No. DE-AC02-05CH11231. Dr. Binshen Zhang and Yiming Niu from Shenyang National Laboratory for Materials Science are appreciated for their kind help in TEM analysis.

## Author contributions

W.X. conceived the research and, with M.M., directed research efforts. F.Y. contributed to experiments and data analysis. A.O.E. and R.S. performed the DFT calculations and mechanistic analysis. P.S. did additional mechanistic analysis. S.Y., R.D., Y.L., and S.S. contributed to the TEM analysis. J.W. and Z.P. contributed to the in situ Raman analysis. The manuscript was primarily written by W.X., F.Y., A.O.E., R.S. and M.M. All authors contributed to discussions and manuscript review.

## Competing interests

The authors declare no competing interests.
