## [Peer Review File · Nature Communications]

Reviewers' comments:

Reviewer #1 (Remarks to the Author):

This manuscript described a new 2D material Bismuthene for producing formate with high selectivity (FE > 99%) and low overpotential by electrochemical reduction of CO₂. DFT simulation and calculation was also conducted to study the mechanism and thermodynamics. The authors claimed that the thickness and structure of BiNS affected the selectivity, and thinner BiNS performed better due to Surface compressive strain. Monolayer of BiNS was claimed.

It also showed that small amount of carbon black (3wt%) increased current density from 3 to 54 mA cm⁻².

Although the results seemed interesting, there were few points the authors should pay attention to. Some of the results did not make sense.

1. Without carbon support, the current was only 3mA cm⁻², which was not significant; with 3wt% carbon, it was increased one magnitude. The authors mentioned that carbon black BP used was not active for CO₂ to formate. This is something in doubt.

Carbon black or activated carbon are known to be active for formate production, and the FE could be around 30%. In S20, the figure showed a more positive shift from CO₂ to N₂. This was strange. In KHCO₃ buffer, when purge N₂, the pH would increase; whereas purging CO₂ would cause pH to decrease from the carbonate system equilibrium. Therefore, it would always be the case, LSV of CO₂ would have a positive shift from N₂ LSV, which was opposite to what the authors presented. These results should be repeated. Error bars should be displayed wherever possible.

2. The calculation of FE in Supplementary information used on S3. Why T₀= 273 K? Shouldn't this be T₀=298K at standard condition?

$$[FE]_{s} = (2Fv_{s} GP_{0}) / (RT_{0} i_{total}) \times 100\%$$

3. On S2, the author stated "an anion-exchange membrane (Nafion 117)". Nafion is a cation exchange membrane!

4. The authors claimed that the catalyst was stable for >75 h. What was the concentration of HCOO⁻ achieved after certain time, for example one hour? What was the HCOO⁻ production rate mM/min cm² or mM/hour cm²?

4. Recently Bi attracted lots of interest for eCO₂RR for formate production, there have been other study claimed. This study <https://www.mdpi.com/1420-3049/24/11/2032>, demonstrated similar results but using carbon supported Bi nanoparticles. Could the authors compare and comment on this work? This method seemed a simpler preparation method.

Reviewer #2 (Remarks to the Author):

Yang and coworkers reported the synthesis of free-standing bismuthene and discovered its high catalytic performance towards CO₂ reduction to formate. Bismuth-based materials (even bulk metal) are very good electrocatalysts for conversing CO₂ to formate in aqueous solution, but with relatively high overpotential requirement. The overpotential for the formation of formate on bismuthene is 100 mV lower than state-of-the-art bismuth-based electrocatalysts, which is quite significant. The synthesis of monolayer bismuthene with a simple wet chemical method is also very

interesting. However, I noticed about several major inconsistencies from the paper. If additional evidence can be provided, this paper may eventually be acceptable for publication. Major comments are listed below:

1. One of my major concerns is lack of strong evidence for formation of the monolayer bismuthene. According to literature, the basal plane of a monolayer bismuthene should show a honeycomb hexagonal arrangement. The authors also claimed that their bismuthene has this hexagonal structure. However, there is no evidence to support their claim. On the contrary, the HRTEM image of the monolayer bismuthene in Fig. 1b shows almost quadrilateral arrangement of Bi atoms.
2. Another major concern is the stability of the bismuthene. As Bi has very high affinity towards O, it can easily be oxidized to form bismuth oxygens. This is especially true for metallic Bi nanomaterials, as they are even more reactive. In aqueous bicarbonate solution, nano metallic Bi can be easily transformed to Bi₂O₃ within minutes, which also has a 2D structure. Therefore, when test the stability of the bismuthene after BE, the authors should also verify the chemical identity of the as obtained 2D nanosheet.
3. The authors claimed that the metallic bismuthene can remain stable at ambient conditions. This is inconsistent with the fact that metallic Bi nanomaterials can be easily oxidized in air. Is the bismuthene been protected by surface surfactant layer? The authors should explain this.
4. Reporting and comparison of current densities with other works is misleading. As far as I can tell, only the geometric current density is reported. Comparison is therefore meaningless when comparing with materials of significantly different morphologies/loadings/supports etc.
5. lines 84: it is surprising that the morphology and thickness remain unchanged when bismuthene is turned into bismuth oxide at 400 °C. Some explanation is needed.
6. Figure 3a: there is clear transition point associated with the voltammogram obtained with bismuthene at the potential of ~-0.8 V. Some explanation is needed.
7. The authors used the value -0.20 vs RHE as the standard equilibrium potential (E₀) for the reduction of CO₂ to formate. This is a wrong value widely used in quite a few recently reported literatures. The right E₀ of this reaction should be pH dependent. Please refer to the following papers for more information: 10.1016/j.nanoen.2016.11.004. I would recommend an E₀ value of -0.09 V vs RHE in CO₂ saturated 0.5M KHCO₃.
8. I noticed that the references in this work is not up to date. CO₂RR is a fast-developing research field. The authors should update their reference list and include those relevant reports published in 2018 and 2019.
9. Please reduce the tick label increment to make the display of Figures 2a, 3a, 3b, 3d, and 4b more readable.
10. the results obtained in this paper, e.g. layer dependent catalytic performance/electrical resistance, are contradictory to those reported by Wang et al (<https://doi.org/10.1021/acs.jpcclett.9b01406>).

Response to the comments

We thank the referees for their time and comments on our manuscript. In the following, we provide a point-by-point response to their comments.

Reviewer #1 (Remarks to the Author):

This manuscript described a new 2D material Bismuthene for producing formate with high selectivity ($FE > 99\%$) and low overpotential by electrochemical reduction of CO_2 . DFT simulation and calculation was also conducted to study the mechanism and thermodynamics. The authors claimed that the thickness and structure of BiNS affected the selectivity, and thinner BiNS performed better due to Surface compressive strain. Monolayer of BiNS was claimed. It also showed that small amount of carbon black (3wt%) increased current density from 3 to 54 $mA\ cm^{-2}$. Although the results seemed interesting, there were few points the authors should pay attention to. Some of the results did not make sense.

1a. Without carbon support, the current was only $3mA\ cm^{-2}$, which was not significant; with 3wt% carbon, it was increased one magnitude. The authors mentioned that carbon black BP used was not active for CO_2 to formate. This is something in doubt. Carbon black or activated carbon are known to be active for formate production, and the FE could be around 30%.

Response: Thanks for your comment. As for the carbon black, indeed it could be active for CO_2RR when the potential is negative enough or when the overpotential is large enough. While for the case here, due to the small overpotential of CO_2RR on Bismuthene nanosheets, as shown in the following Fig. R1a (pH-corrected linear sweep voltammetric curves (LSV), such as *Chem* 2017, 3, 1; *Joule* 2019, 3, 265), we only measured the catalytic activity of BP in the same potential range as shown in the following Fig. R1b. Obviously, compared with the activity of Bismuthene, the activity of BP for CO_2RR could be neglected. To further confirm this point, we supplemented the FE measurement on BP as shown in Fig. R1c. It clearly shows that the small FE for the only liquid product $HCOO^-$ is negligible. Based on this comment, we revised the statement in the revised manuscript (page-5) from "...BP, which is inert to CO_2RR " to "...BP, which is inert to CO_2RR in the potential range studied here".

Fig. R1. (a) pH-corrected linear sweep voltammetric curves (LSV) in the CO_2 saturated (solid line pH 7.2) and N_2 saturated (dash line pH 8.8) 0.5 M $KHCO_3$ aqueous solution with Bi loading of 0.39 mg/cm^2 on glassy carbon electrode. (b) pH-corrected LSV in the CO_2 saturated (pH 7.2) and N_2 saturated (pH 8.8) 0.5 M $KHCO_3$ aqueous solution with BP on glassy carbon electrode. (c) The potential-dependent FE of different products on BP.

1b. In S20, the figure showed a more positive shift from CO₂ to N₂. This was strange. In KHCO₃ buffer, when purge N₂, the pH would increase; whereas purging CO₂ would cause pH to decrease from the carbonate system equilibrium. Therefore, it would always be the case, LSV of CO₂ would have a positive shift from N₂ LSV, which was opposite to what the authors presented. These results should be repeated.

Response: As for the pH-corrected LSV curves of BP in N₂ and CO₂ solution shown in Fig. S20 or in the following Fig. R2a, actually, we have repeated such measurement several times and obtained the same shape as that shown in Fig. R2a. Even before the pH-correction, as shown in Fig. R2b, the same trend was observed, consistent with previous observation by others (*Angew. Chem.* 2018, 130, 9788). To further confirm its reliability, we also measured the same process on Vulcan XC-72 carbon powder, as shown in Fig. R2c-f, interestingly, it shows the same trend as BP shown in Fig. R2a,b. Actually, similar observations have been made extensively by other labs (such as *Angew. Chem.* 2018, 130, 9788; *ACS Catal.* 2015, 5, 3916; *Chem* 2017, 3, 1; *J. CO₂ Utilization* 2017, 18, 41; *Nat. Commun.* 2013, 4, 2819; *J. Am. Chem. Soc.* 2017, 139, 8078) as shown in Fig. R3 (for pure carbon materials) and Fig. R4 (for carbon-supported catalysts).

Fig. R2 (a) Before and after (b) pH-correction of LSVs in the CO₂ saturated (pH 7.2) and N₂ saturated (pH 8.8) 0.5 M KHCO₃ aqueous solution with BP on glassy carbon electrode. (c) Before and after (d) pH-correction of LSVs in the CO₂ saturated (pH 7.2) and N₂ saturated (pH 8.8) 0.5 M KHCO₃ aqueous solution with XC-72 on glassy carbon electrode. (e) Before and after (f) pH-correction of CVs in the CO₂ saturated (pH 7.2) and N₂ saturated (pH 8.8) 0.5 M KHCO₃ aqueous solution with XC-72 on glassy carbon electrode.

Fig. R3 Typical results reported by others about the LSV or CV curves of different types of carbon materials in N₂- or CO₂-saturated solution.

Fig. R4 Typical results reported by others about the LSV curves of different metal catalysts supported by carbon materials (such as Vulcan XC-72) in N₂- or CO₂-saturated solution. These catalysts are all active for CO₂RR process according their observation.

2. The calculation of FE in Supplementary information used on S3. Why T₀= 273 K? Shouldn't this be T₀=298K at standard condition?

Response: Thanks for your reminder here. We double-checked our data analysis and found that, in our FE calculation, the value of 298.15 K rather than 273.15 K was adopted for T₀. Hence, T₀= 273 K was a typo, and all the values of FE presented in this manuscript are reliable. We made the corresponding revision about the value of T₀ in SI.

3. On S2, the author stated “an anion-exchange membrane (Nafion 117)”. Nafion is a cation exchange membrane!

Response: Thanks for your reminder here. Based on it, we have done the corresponding revision properly by changing the word “anion” to “cation”.

4. The authors claimed that the catalyst was stable for >75 h. What was the concentration of HCOO⁻ achieved after certain time, for example one hour? What was the HCOO⁻ production rate mM/min cm² or mM/hour cm²?

Response: We have supplemented the calculation of concentration (mM) of HCOO⁻ achieved after different times of CO₂RR on Bismuthene@BP as shown in Fig. R5a. However, due to the differences of solution volumes and the sizes of electrodes adopted in different labs, for a convenient comparison among different catalysts from different labs, the community usually adopts the product formation rate in "mmol/min cm²" or "mmol/hour cm²" rather than "mM/hour cm²" to characterize the catalytic activities of their catalysts (such *Angew. Chem. Int. Ed.* 2016, 55, 9297; *Nano Energy* 2017, 39, 44; *Nat. Commun.* 2016, 7, 13869). Here, prompted by your question, we also supplemented the calculation of the HCOO⁻ production rate in "mmol/hour cm²" in the revised manuscript (page-5, Fig. S20b) and in Fig. R5b.

Fig. R5 (a) Formate concentration vs time at different potentials on Bismuthene@BP. (b) Formate production rates at different potentials on Bismuthene and Bismuthene@BP.

5. Recently Bi attracted lots of interest for eCO₂RR for formate production, there have been other study claimed. This study <https://www.mdpi.com/1420-3049/24/11/2032>, demonstrated similar results but using carbon supported Bi nanoparticles. Could the authors compare and comment on this work? This method seemed a simpler preparation method.

Response: Thanks for the information you shared. Based on your suggestion, we read this paper (*Molecules* 2019, 24, 2032) carefully and then made a simple comparison as shown in the following. Firstly, their method is indeed simpler than the one reported here; secondly, the obtained monolayer Bismuthene in our work presents much higher onset potential or much smaller overpotential for the formate formation from CO₂RR; thirdly, compared with their carbon supported Bi nanoparticles (about 10 nm in size), the durability of Bismuthene@BP on electrode is higher. We have now included a citation to this paper in the revised manuscript: Ref.15.

Reviewer #2 (Remarks to the Author):

Yang and coworkers reported the synthesis of free-standing bismuthene and discovered its high catalytic performance towards CO₂ reduction to formate. Bismuth-based materials (even bulk metal) are very good electrocatalysts for conversing CO₂ to formate in aqueous solution, but with relatively high overpotential requirement. The overpotential for the formation of formate on bismuthene is 100 mV lower than state-of-the-art bismuth-based electrocatalysts, which is quite significant. The synthesis of monolayer bismuthene with a simple wet chemical method is also very interesting. However, I noticed about several major inconsistencies from the paper. If additional evidence can be provided, this paper may eventually be acceptable for publication. Major comments are listed below:

1. One of my major concerns is lack of strong evidence for formation of the monolayer bismuthene. According to literature, the basal plane of a monolayer bismuthene should show a honeycomb hexagonal arrangement. The authors also claimed that their bismuthene has this hexagonal structure. However, there is no evidence to support their claim. On the contrary, the HRTEM image of the monolayer bismuthene in Fig. 1b shows almost quadrilateral arrangement of Bi atoms.

Response: To address this comment, we present here the relationship between the structure, as determined from HRTEM images, and optimized geometries obtained from DFT calculations, shown in Fig. R6. In order to describe the zig-zag arrangement for Bi atoms in the Bi(111) facet (Fig. S24a in SI), the atoms in the first layer were colored in blue, and those in the second layer in purple (Fig. R6a,b). When the structure was rotated 45° (Fig. R6c) and 90° down (Fig. R6d), the hexagonal pattern formed by the Bi atoms in the first and second layer is evident. Actually, they form a typical hexagonal chair structure in 3D space (Fig. R6f). In the HRTEM image (Fig. R6e), we find a distribution of bright spots, corresponding to Bi atoms in the first layer, around dark spots, corresponding to Bi atoms in the second layer. This specific pattern forms the zig-zag structures in Fig. R6a. Based on such features, we can also find a hexagon-like projection (the green one in the inset of Fig. R6e), and the real three-dimensional structure is chair-like hexagonal, as shown in Fig. R6f. So, for the Bismuthene monolayer obtained here, the real space configuration is indeed a honeycomb hexagonal arrangement, while in HRTEM, due to the fact that the upper atoms are projected exactly on either side of the lower atoms, a quadrilateral-like arrangement of Bi atoms could be observed apparently in 2D HRTEM image. Such analysis has been supplemented to the revised SI as Fig. S3.

Fig. R6 The structure arrangement for Bi(111) ((a) and (b)), and rotated 45° (c) and 90° down (d); (e) HRTEM image; the honeycomb hexagonal arrangement in yellow area is reported in the inset; (f) Space configuration for the honeycomb hexagonal arrangement in Figure R6e; (g) Honeycomb hexagonal arrangement (green ring) in the optimized geometry, at the PBE+TS level, of a Bismuthene monolayer.

2. Another major concern is the stability of the bismuthene. As Bi has very high affinity towards O, it can easily be oxidized to form bismuth oxyanion. This is especially true for metallic Bi nanomaterials, as they are even more reactive. In aqueous bicarbonate solution, nano metallic Bi can be easily transformed to BiOCO_3 within minutes, which also has a 2D structure. Therefore, when test the stability of the bismuthene after BE, the authors should also verify the chemical identify of the as obtained 2D nanosheet.

Response: In this case, firstly, we observed the oxidation of Bi nanosheets when they are heated at 400 °C in air (Fig. S7 in SI and Fig. R7 shown below). However, we found that the metallic Bi nanosheets can be maintained stable for several months, when stored in an airtight Ar atmosphere at 4 °C.

Fig. R7 (a) Raman spectrum and (b) high resolution 4f XPS spectrum of annealed Bismuthene nanosheets. It shows the formation of Bi oxide after such annealing process in air.

Moreover, as for the stability of such Bi nanosheets in aqueous bicarbonate solution, we also analyzed their XRD and XPS spectra before and after long-term (~ 75 h) CO₂RR at -0.58 V. As shown in the Fig. R8, or Fig. 2a in the manuscript, one can see clearly that there is almost no oxidation of Bismuthene nanosheets after CO₂RR in aqueous bicarbonate solution. This evidence confirms the stability of Bismuthene nanosheets and no formation of bismuth carbonate in this case during CO₂RR in aqueous bicarbonate solution. Actually, it has been known that the oxidation of Bi metal is the prerequisite for the formation of Bi₂O₂CO₃ in aqueous bicarbonate solution (*Angew. Chem. Int. Ed.* 2018, 57, 13283). In this case, due to the facts that metallic Bismuthene can be stored well in Ar atmosphere for long time and the CO₂RR process usually occurs at negative potentials (such as -0.58 V), the oxidation of metallic Bismuthene can be prevented well during CO₂RR process in aqueous bicarbonate solution. Without the formation of Bi oxide, then the formation of Bi₂O₂CO₃ cannot occur in aqueous bicarbonate solution (*Angew. Chem. Int. Ed.* 2018, 57, 13283). The relevant paper is now cited in the manuscript as Ref. 21.

Fig. R8 High resolution Bi⁰ 4f XPS spectra of metallic Bismuthene nanosheets before/after CO₂RR process in aqueous bicarbonate solution.

3. The authors claimed that the metallic bismuthene can remain stable at ambient conditions. This is inconsistent with the fact that metallic Bi nanomaterials can be easily oxidized in air. Is the bismuthene been protected by surface surfactant layer?

The authors should explain this.

Response: In this work, the obtained metallic Bi nanosheets were stored in an airtight Ar atmosphere at 4 °C in a refrigerator. In this way, metallic Bi nanosheets can be stored stable for several months. Actually, as the referee expected, when the sample was exposed to air, it indeed got oxidized gradually as our control experiments show in the following Fig. R9.

We did not add any surfactant to the solution for synthesis, so there is no surfactant on the surface of Bismuthene. Such fact has been confirmed by the EDS analysis on individual metallic Bismuthene nanosheets (Fig. S6 in SI).

Fig. R9 High resolution 4f XPS spectrum of fresh Bismuthene nanosheets and the Bismuthene after exposure to air for one week and two weeks, respectively.

4. Reporting and comparison of current densities with other works is misleading. As far as I can tell, only the geometric current density is reported. Comparison is therefore meaningless when comparing with materials of significantly different morphologies/loadings/supports etc.

Response: We checked our manuscript thoroughly and confirmed that we did not make any comparison of current densities of our catalysts with works reported by others. Notice that we only compare the peak potentials and FE, but not current densities (see Table S1). The only place in the manuscript we mentioned current density was in the context of durability of the catalyst as shown in the following sentence (page-5): "...in a long time (75 h) window, the current density and the corresponding FE_{HCOO} were maintained steadily at around -3.0 mA cm^{-2} and 98%, respectively, indicating a remarkable durability....". Given your comment, and to avoid any confusion, we revised that sentence as follows: "...in a long time (75 h) window, both the current density and the corresponding FE_{HCOO} were maintained steadily, indicating a remarkable durability....".

5. lines 84: it is surprising that the morphology and thickness remain unchanged when bismuthene is turned into bismuth oxide at 400 °C. Some explanation is needed.

Response: In this work, we indeed observed the stable morphology (2D layered structure) and the slight variation of thickness (from $0.65 \pm 0.05 \text{ nm}$ to $0.68 \pm 0.07 \text{ nm}$) of Bismuthene before/after annealing treatment. These facts indicate that, after the oxidation of bismuthene, a monolayer oxide (see Fig. S5c,d and Fig. R7) was obtained. Moreover, based on the metallic Bi monolayer structure revealed (Fig. 1b,c) and the previous knowledge about the Bi oxide (*J. Phys.: Condens. Matter* 2013, **25**, 475402), we estimated roughly the thickness of the monolayer Bi oxide as shown in the following Fig. R10. It shows that the thickness of the layered structure varies slightly

from 0.51 nm to 0.69 nm after oxidation. Such slight difference is consistent with the AFM results shown in Fig. S5. The obtained primary new results indicate that the embedding of oxygen atom can slightly increase the thickness of the layered structure, but does not change the orientation of the Bi atoms in the structure and then can maintain its monolayer morphology. In our future study, we will do more detailed work from both experimental and theoretical points of views to understand deeply such observation about the stable morphology and thickness.

Fig. R10 Structures for monolayer Bismuthene (a) and bismuth oxide (b) for thickness estimation.

6. Figure 3a: there is clear transition point associated with the voltammogram obtained with bismuthene at the potential of ~ -0.8 V. Some explanation is needed.

Response: We thank the reviewer for raising this point. The “transition point” you mentioned is difficult to rationalize without a more in depth analysis, which is beyond the scope of this work. While we probably could ascribe the transition point to a HER-induced fast current increase. This can be inferred by the abrupt increase of FE_{H_2} at potential lower than -0.78 V, as highlighted in Fig. R11 (marked by the red arrow) and Fig. S9B. We plan to study this point deeply in the future.

Fig. R11 FEs of H_2 at various applied potentials on Bismuthene nanosheets.

7. The authors used the value -0.20 vs RHE as the standard equilibrium potential (E_{eq}) for the reduction of CO_2 to formate. This is a wrong value widely used in quite a few recently reported literatures. The right E_{eq} of this reaction should be pH dependent. Please refer to the following papers for more information: 10.1016/j.nanoen.2016.11.004. I would recommend an E_{eq} value of -0.09 V vs RHE in CO_2 saturated $0.5M$ $KHCO_3$.

Response: Thanks for your reminder here. Based on it, we double checked our data analysis, and found that the reported "onset overpotential of < 180 mV" in the old manuscript was actually based on the point marked by a red circle shown in the following Fig. R12 or Fig. S9b, and the value of $E_{eq}^0 = 0.0$ V rather than -0.20 V for

the reduction of CO₂ to formate was adopted, as indicated by the sentence at the end of Table S1: "E⁰ (CO₂/HCOO⁻) = 0 V (vs. RHE in neutral solutions)" in the old SI (see the following screen shot (Fig. R13)). In this revision, based on your recommendation, we have recalculated the values of overpotential based on the value of -0.09 V you recommended. The paper you mentioned and the related one (*Energy Environ. Sci.* 2019, 12, 1334) have also been cited as Ref. 18,36 in the revised manuscript.

Fig. R12 The screen shot of Fig. S9b in SI to show the calculation of the onset overpotential of CO₂RR from the point marked with red circle at -0.18 V.

reduced mpBi nanosheets ^a	0.5 M NaHCO ₃ ^a	-0.9 V vs.RHE ^a	99% ^a	Ref. S30 ^a
Cu foam@BiNW ^a	0.5 M NaHCO ₃ ^a	-0.69 V vs.RHE ^a	95% ^a	Ref. S31 ^a
Bi ₂ O ₃ -NGQDs ^a	0.5 M KHCO ₃ ^a	-0.9 V vs.RHE ^a	100% ^a	Ref. S32 ^a
Bi ₂ O ₃ NSs@MCCM ^a	0.1 M KHCO ₃ ^a	-1.256 V vs.RHE ^a	93.8% ^a	Ref. S33 ^a
POD-Bi ^a	0.5 M KHCO ₃ ^a	-1.16 V vs.RHE ^a	95% ^a	Ref. S34 ^a
E ⁰ (CO ₂ /HCOO ⁻) = 0 V (vs. RHE in neutral solutions). ^a				

Fig. R13 The screen shot to show the sentence at the end of Table S1 in the old SI to indicate the adoption of 0.0 V as the E⁰ (CO₂/HCOO⁻).

8. I noticed that the references in this work is not up to date. CO₂RR is a fast-developing research field. The authors should update their reference list and include those relevant reports published in 2018 and 2019.

Response: Thanks for your reminder here. Based on it, we have updated the references by supplementing the citation of 9 new papers published recently in the revised manuscript: Ref.6, Ref.11, Ref.12, Ref.13, Ref.15, Ref.18, Ref.21, Ref.28, Ref.36.

9. Please reduce the tick label increment to make the display of Figures 2a, 3a, 3b, 3d, and 4b more readable.

Response: Thanks for your reminder here. Based on it, we have done the corresponding revision carefully for all the figures in the manuscript.

10. the results obtained in this paper, e.g. layer dependent catalytic performance/electrical resistance, are contradictory to those reported by Wang et al (<https://doi.org/10.1021/acs.jpcclett.9b01406>).

Response: We recognize the apparent discrepancy highlighted by the reviewer. However, the computational work of Wang et al. is focused on Bi(0001), while in our own study, we make a comparison between a Bi(111) monolayer and a Bi(011) slab.

Hence, the comparison is not only across different thicknesses, but also across different facets. If we eliminate facet differences, and focus only on thickness effects for a given facet, we observe similar trends to those reported by Wang et al. For Bi(111), we confirm that Bi multilayers are more active than a Bi monolayer, in agreement with Wang et.al. In the following Fig. R14, we show our calculations on the CO₂ reduction reaction (CO₂RR) on a Bi monolayer and a Bi(111) slab composed of 9 layers.

Fig. R14 Calculated free energy diagrams for CO₂RR and HER on (111) single-atom-thick Bismuthene (continuous lines) and a 9-layer (111) slab (dashed lines), at 0.0 V.

In agreement with Wang et al., we find a decrease in the overpotential for the CO₂RR as the Bi surface thickness increases. We ascribe this behavior to the more compressive strain experienced by the monolayer with respect to the thicker, 9-layer slab. The change in binding strength with thickness could also be ascribed to the band-gap variation, as suggested by Wang et al.

The SAED measurements reported in our manuscript suggest that Bismuthene nanosheets expose the (111) facet, while thicker nanosheets expose the (011). The difference in reactivity leading to the conclusion “the thinner, the better” is given by the structure sensitivity of CO₂RR on the (011) multi-layered nanosheets and (111) monolayers, more than counterbalancing the thickness trends observed by us (Figure 1) and Wang et al.

Based on your reminder, such paper has been cited in the manuscript as Ref.28.

Reviewers' comments:

Reviewer #1 (Remarks to the Author):

For this revision, I think the authors addressed the issues, therefore, I am happy to support the manuscript to be published.

Reviewer #3 (Remarks to the Author):

Yang et al. reported the preparation of bismuthene for carbon dioxide electroreduction to formate. It is found that the thickness and the large compressive strain affect catalytic performance significantly. This work presents some interesting results, and is acceptable for publication after major revision.

1. Bi is a very common metal for CO₂ reduction. The main advantages and disadvantages of present study should be added in the introduction part.
2. The current density at typical condition should be provided in the Abstract.
3. In the abstract, it is said that "Bismuthene and demonstrate its superhigh electrocatalytic efficiency for formate (HCOO⁻) formation". This is an overstatement because the current density is very common.
4. TEM images in large scale of Bi nanosheets should be shown in Fig. 1.
5. Carbon black was used as support to enhance the current density. The morphology changed a lot as well as the Bi content, which may affect the activity. Therefore, the factors to influence the reaction performance should be discussed in detail.
6. An equivalent circuit should be used to fit the data of Nyquist plots.
7. CO₂ temperature-programmed desorption seems unreasonable in CO₂ electroreduction. The reaction conditions are quite different (high temperature vs. room temperature). The authors should give more evidences to explain the CO₂ chemisorption.
8. Due to the chemical unstability of bismuthene, how Bi changes in the reaction process? Some operando measurements can be performed.
9. How about XPS spectra of Bi after electrolysis of 75 h? Was the chemical valence remains unchanged?
10. How stable is the matrix structure? Especially under long-term high current density operation

Response to the comments

Reviewers' comments:

Reviewer #1 (Remarks to the Author):

For this revision, I think the authors addressed the issues, therefore, I am happy to support the manuscript to be published.

Response: Dear Reviewer, many thanks for your time and comments which helped a lot for the improvement of this work.

Reviewer #3 (Remarks to the Author):

Yang et al. reported the preparation of bismuthene for carbon dioxide electroreduction to formate. It is found that the thickness and the large compressive strain affect catalytic performance significantly. This work presents some interesting results, and is acceptable for publication after major revision.

Response: Dear Reviewer, we thank you very much for your time and constructive comments on our manuscript. In the following, by supplementing more discussion and new data, we provide a point-by-point response to your comments.

1. Bi is a very common metal for CO₂ reduction. The main advantages and disadvantages of present study should be added in the introduction part.

Response: Thanks for your reminder. Based on it, in this revision, we have supplemented the following sentences to the introduction part (page-2 in the revised manuscript): “Bismuth (Bi), typically used as CO₂RR electrocatalyst, has advantages over other traditional metals due to its low toxicity, cost-effectiveness, and higher stability. However, previously reported bismuth electrocatalysts for CO₂RR usually require large overpotentials or only present low current densities.” About the advantage of our work, it can be found in the following sentence in page-2: “Here we report a simple, scalable, wet chemical method to synthesize stable free-standing hexagonal Bismuthene and demonstrate its superior electrocatalytic CO₂RR performance towards HCOO⁻ production.”

2. The current density at typical condition should be provided in the Abstract.

Response: Many thanks for your reminder here. Based on your suggestion, we have provided the current density at typical condition in the abstract.

3. In the abstract, it is said that “Bismuthene and demonstrate its superhigh electrocatalytic efficiency for formate (HCOO⁻) formation”. This is an overstatement because the current density is very common.

Response: Many thanks for your comment here. Based on it, we have revised the word of "superhigh" to "remarkable".

4. TEM images in large scale of Bi nanosheets should be shown in Fig. 1.

Response: Thanks for your comment. Based on it, we have provided two TEM images of monolayer Bi nanosheets in large scale as shown in Fig.1a,b in the revised manuscript or in the following Fig. R1.

Fig. R1 Typical TEM images of monolayer Bi nanosheets in large scale.

5. Carbon black was used as support to enhance the current density. The morphology changed a lot as well as the Bi content, which may affect the activity. Therefore, the factors to influence the reaction performance should be discussed in detail.

Response: Many thanks for your comment. Based on it, we supplemented more discussion about the performance improvement after the addition of carbon black: “In all, such improvement of CO₂RR performance could be attributed in part to the BP-induced variations of morphology (compact & noncompact), Bi content and ECSA of the catalyst layer and then the variation of mass transportation in it.” More details can be found at page-6 of the revised manuscript.

6. An equivalent circuit should be used to fit the data of Nyquist plots.

Response: Thanks for your comment. Based on it, we have supplemented the fits to the revised manuscript. More details can be found in Fig. 3e and Fig. S12.

7. CO₂ temperature-programmed desorption seems unreasonable in CO₂ electroreduction. The reaction conditions are quite different (high temperature vs. room temperature). The authors should give more evidences to explain the CO₂ chemisorption.

Response: Many thanks for your comment. It should be noted here, to obtain the CO₂ temperature-programmed desorption (TPD) spectrum as shown in Fig. S11, CO₂ molecules have to be adsorbed on Bi nanosheets first at room temperature (25 °C); after that, the sample was heated up at a rate of 5 °C/min in He atmosphere to record the desorption spectrum during the temperature-programmed heating process. Such TPD spectra have been adopted extensively to investigate the CO₂ adsorption capabilities of many kinds of samples (such as *Angew. Chem. Int. Ed.* 2018, 57, 10954; *ACS Catal.* 2019, 9, 336). For the case here, the 0.65 nm-nanosheets (Fig. S11 or the

following Fig. R2a) exhibit the largest CO₂ desorption peak area, followed by 4.2 nm and 11.3 nm, consistent with CO₂ adsorption isotherms of BiNSs with different thickness (Fig. R2b), all indicating that the monolayer Bismuthene has the largest CO₂ adsorption capability. Based on your reminder, we have made the corresponding revision in the manuscript (the content below Fig. S10 in Page-S10).

Fig. R2 (a) CO₂-TPD spectra and (b) CO₂ adsorption isotherms of BiNSs with different thicknesses.

8. Due to the chemical unstability of bismuthene, how Bi changes in the reaction process? Some operando measurements can be performed.

Response: Many thanks for your comment. Based on it, we supplemented an *in-situ* Raman analysis to monitor the stability of Bismuthene during CO₂RR process based on the two characteristic Raman bands at 74 and 99.7 cm⁻¹, corresponding to the E_g and A_{1g} modes of metallic Bi (0). The following *in-situ* surface enhanced Raman spectroscopic (SERS) (Fig. R3) show, at fixed electrode potential of -0.58 V vs. RHE, the SERS signal is consistent with the Raman signal obtained from the dry powder of fresh metallic Bismuthene shown in Fig. 2b. There is no variation over a long time window. All these facts along with the stable morphology observed on TEM images (Fig. S18) indicate that the monolayer Bismuthene nanosheets are indeed stable during the CO₂RR process.

Based on your comment here, such new data were supplemented to SI as Figure S18.

Fig. R3 (a) Scheme of *in-situ* SERS set-up and the spectroelectrochemical cell. (b) Real-time *in-situ* Raman spectroscopic of Bismuthene in a CO₂-saturated 0.5 M KHCO₃ solution at -0.58 V vs. RHE.

9. How about XPS spectra of Bi after electrolysis of 75 h? Was the chemical valence remains unchanged?

Response: Thanks for your comment. We have already presented XPS spectrum of Bismuthene nanosheets before/after 75-hrs CO₂RR process at -0.58 V in the manuscript (Fig. 2a). It shows clearly that the chemical valence remains unchanged after electrolysis of 75 h.

10. How stable is the matrix structure? Especially under long-term high current density operation.

Response: As for the stability of matrix structure you mentioned, to make it clear, we supplemented the HRTEM analysis of the monolayer Bismuthene after long-term (75 hrs) high current density operation (at fixed potential of -0.88 V vs. RHE, the most negative potential we tested in Fig. S14). As shown in the following Fig. R4, after long-term high current density operation, the lattice matrix structure shows almost no change. Such fact further confirms the remarkable durability of monolayer Bismuthene during the CO₂RR process, just like that shown by TEM (Fig. S19) and Raman (Fig. S17-S18) analysis.

Based on your comment here, we have supplemented such new result to the revised SI as Fig. S21.

Fig. R4 HRTEM-based structural analysis of Bismuthene nanosheets before (a) and after (b) long-term (75 hrs) high current density operation.

REVIEWERS' COMMENTS:

Reviewer #3 (Remarks to the Author):

In this resubmitted manuscript, the authors have answered our questions systematically. This manuscript is suggested to be accepted.

Response to the Comments

REVIEWERS' COMMENTS:

Reviewer #3 (Remarks to the Author):

In this resubmitted manuscript, the authors have answered our questions systematically. This manuscript is suggested to be accepted.

Response: Dear Reviewer, many thanks for your comments and time on our manuscript. You helped a lot for the improvement of this work. We appreciate it very much.